# 4,6,4′-Trimethylangelicin Photoactivated by Blue Light Might Represent an Interesting Option for Photochemotherapy of Non-Invasive Bladder Carcinoma: An In Vitro Study on T24 Cells

**DOI:** 10.3390/biom11020158

**Published:** 2021-01-25

**Authors:** Giulio Sturaro, Alessia Tasso, Luca Menilli, Rosa Di Liddo, Giorgia Miolo, Maria Teresa Conconi

**Affiliations:** Department of Pharmaceutical and Pharmacological Sciences, University of Padua, via Marzolo, 5, 35131 Padova, Italy; giulio.sturaro@gmail.com (G.S.); alessia.tasso@unipd.it (A.T.); luca.menilli@unipd.it (L.M.); rosa.diliddo@unipd.it (R.D.L.); mariateresa.conconi@unipd.it (M.T.C.)

**Keywords:** bladder cancer, T24 cell line, 4,6,4′-trimethylangelicin, Wnt signalling pathway

## Abstract

Photodynamic therapy (PDT) is frequently used to treat non-muscle invasive bladder cancer due its low toxicity and high selectivity. Since recurrence often occurs, alternative approaches and/or designs of combined therapies to improve PDT effectiveness are needed. This work aimed to evaluate the cytotoxicity of 4,6,4′-trimethylangelicin (TMA) photoactivated by blue light (BL) on human bladder cancer T24 cells and investigate the mechanisms underlying its biological effects. TMA/BL exerted antiproliferative activity through the induction of apoptosis without genotoxicity, as demonstrated by the expression levels of phospho-H2AX, an indicator of DNA double-stranded breaks. It also modulated the Wnt canonical signal pathway by increasing the phospho-β-catenin and decreasing the nuclear levels of β-catenin. The inhibition of this pathway was due to the modulation of the GSK3β phosphorylation state (Tyr 216) that induces a proteasomal degradation of β-catenin. Indeed, a partial recovery of nuclear β-catenin expression and reduction of its phosphorylated form after treatment with LiCl were detected. As demonstrated by RT-PCR and cytofluorimetric analysis, TMA/BL also decreased the expression of CD44v6, a marker of cancer stem cells. Taken together, our data suggest that TMA photoactivated by BL may represent an interesting option for the photochemotherapy of noninvasive bladder carcinomas, since this treatment is able to inhibit key pathways for tumour growth and progression in the absence of genotoxic effects.

## 1. Introduction

Nowadays, non-muscle invasive bladder cancer is treated with transurethral resection followed by intravesical chemotherapy or immunotherapy to ablate unresected tumour tissue and floating cancer cells, which are responsible for relapsing. In photodynamic therapy (PDT), frequently used when standard intravesical agents fail, visible light activates a photosensitiser, leading to the production of reactive oxygen species (ROS) that induce cell death. 5-aminolevulinic acid (ALA) and its hexyl ester (hALA) act as prodrugs, because they are converted into protoporphyrin IX (PpIX), having photosensitising properties [1]. They also show red fluorescence (600–740 nm) useful to detect tumours and guide surgical resection through irradiation with visible blue light (375–445 nm) [2]. The main advantages of ALA-based PDT are low toxicity and high selectivity towards tumour cells. Indeed, thanks to alterations in heme biosynthetic enzymes, porphyrin transporters, and mitochondrial function, PpIX accumulates in tumour cells at a higher extent than in normal ones [3]. Nevertheless, long-term recurrence and progression of bladder cancer is often observed in patients, mainly due to cell resistance against ROS damage [4]. Thus, alternative therapeutic approaches and/or the design of combined therapies to improve PDT effectiveness are needed.

Furocoumarins are natural and synthetic compounds, which include both linear and angular molecules, called psoralens and angelicins, respectively. 8-methoxypsoralen (8-MOP) is a furocoumarin commonly combined with UVA light (365 nm) in PUVA (psoralen and ultraviolet A) therapy to treat various skin diseases, such as psoriasis, vitiligo, and cutaneous T-cell lymphoma. In the dark, furocoumarins lack cytotoxicity, whereas, under UVA irradiation, they switch to excited states and exert cytotoxic effects through various mechanisms. Photoactivated furocoumarins can bind DNA with the formation of monoadducts and inter-strand crosslinks, the latter entailing the risk of induced mutagenicity. They are also able to photo-bind lipids and intra- and extracellular proteins, leading to mitochondria damage-mediated cell apoptosis [5]. Furthermore, they produce ROS that impair cell functions through lipid peroxidation, the oxidation of guanine, strand breaks in nucleic acids, the oxidation of proteins, and the inactivation of enzymes. The photoactivation of furocoumarins yields stable cytotoxic species (photoproducts), mainly formyl-derivatives and dimers, which are considered to play a relevant role in PUVA therapy [6].

Several evidences indicate that 4,6,4′-trimethylangelicin (TMA), an angular furocoumarin (Figure 1), can also interact with molecular targets under UVA light without photoactivation: it inhibits nuclear factor kappa-light-chain-enhancer of activated B cells (NF-kB)/DNA interactions, thus exerting an anti-inflammatory activity [7], and enhances the Cystic Fibrosis Transmembrane Conductance Regulator (CFTR) [8].

We recently demonstrated, for the first time, that TMA can be also activated by blue light (BL) (see the molar extinction coefficient at 417 nm in Figure 1), exerting strong antiproliferative effects on DU145 prostate cancer cells without the induction of photocleavage and photooxidative damages on DNA and crosslink formations, thus decreasing the potential risk of mutagenic lesions on treated cells [9]. Besides ROS generation, the proapoptotic effects on DU145 cells seemed to be related to the activation of p38 and inhibition of p44/42 phosphorylation. Interestingly, photoactivated TMA was also able to affect Wnt signalling and CD44 expression, both involved in cancer stem cell (CSC) growth and renewal.

Starting from these bases, this work aimed to evaluate the cytotoxicity of TMA photoactivated by BL (TMA/BL) on human bladder cancer cells and investigate the mechanisms underlying its biological effects.

## 2. Materials and Methods

### 2.1. Cell Cultures

Human bladder cancer T24 cells (ATCC, Manassas, VA, USA) were cultured with a proliferation medium composed of Eagle’s Minimum Essential Medium (MEM) (Sigma-Aldrich, Saint Louis, MO, USA) supplemented with 10% fetal bovine serum (FBS), 1% L-glutamine, 0.1-mM nonessential amino acids, and 1% penicillin/streptomycin (Sigma-Aldrich). Cells were grown at 37 °C in a humified atmosphere with 5% CO_2_.

### 2.2. Cell Treatments

Cells (5 × 10^4^) were seeded in 35-mm Petri dishes (Sarstedt, Nϋmbrecht, Germany). After 24 h, the proliferation medium was removed and replaced with MEM w/o phenol red (Sigma-Aldrich) containing various concentrations of TMA (ranging from 1 to 10 µM), kindly provided by Prof. Adriana Chilin (Department of Pharmaceutical and Pharmacological Sciences, University of Padova, Padova, Italy). Alternatively, the medium was added with 1-mM LiCl (Sigma-Aldrich). After 1 h of incubation in the dark at 37 °C, cultures were irradiated with 2 J/cm^2^ BL by using the Waldmann UV436HF apparatus (Herbert Waldmann GmbH & Co. KG, Villingen-Schwenningen, Germany) mainly emitting at about 420 nm. The total energy hitting the sample was monitored by means of a radiometer (Variocontrol, Waldmann), equipped with a Variocontrol UV Sensor (Waldmann). The radiant power emitted by the BL lamp was about 20 mW/cm^2^. The samples were maintained at room temperature (RT) during irradiation. Then, media were removed and replaced with the proliferative one. Nontreated cultures were taken as the control. Evaluation of cell viability and apoptosis and expression studies were carried out at 24 and 48 h after irradiation.

### 2.3. Cell Viability and Apoptosis

Cells were detached, and cell viability was determined by trypan blue staining. Alternatively, cell suspensions were treated with fluorescein isothiocyanate (FITC)-Annexin V and propidium iodide (BD Biosciences, San Jose, CA, USA) and incubated for 15 min in the dark at room temperature (RT). Data were collected with a BD FACSCanto II cytofluorimeter (BD Biosciences, San Jose, CA, USA). Samples marked with only FITC-Annexin V or propidium iodide, and nonstained cells were used as the control to set up the instrument.

### 2.4. Immunofluorescence

Air-dried slides of cells were prepared using Cytospin 4 (Thermo Fisher Scientific, Inc., Waltham, MA, USA) at 1500 rpm for 3 min. The samples were fixed in 10% buffered formalin (Sigma-Aldrich) for 15 min at 4 °C. To detect β-catenin, permeabilisation was carried out through treatment with 0.5% Triton X-100 for 15 min at RT. The inactivation of nonspecific binding sites was achieved by incubation with 3% bovine serum albumin (BSA; Sigma-Aldrich) in phosphate buffer (PBS) for 1 h at RT. Then, slides were incubated overnight at 4 °C with primary nonconjugate mouse monoclonal antibodies: anti-CD44 (AbD Serotech, Kidlington, UK) or anti-β-catenin (Santa Cruz Biotechnology, Dallas, TX, USA) antibodies diluted 1:100 and 1:200 in 1% BSA, respectively. Then, after, samples were incubated for 30 min at RT with secondary antibody goat anti-mouse Alexa 488 (Thermo Fisher Scientific) diluted 1:200 in 1.5% BSA. Slides were mounted with Fluoroshield with 4′,6-diamidino-2-phenylindole (DAPI) (Vector Laboratories, Peterborough, UK) mounting medium. Otherwise, samples were treated with propidium iodide (50 μg/mL, Sigma-Aldrich) for 5 min and then mounted with FLUORO-GEL (Thermo Fisher Scientific).

### 2.5. Flow Cytometry

Cells were detached, washed with 0.5% BSA in PBS, and treated with primary nonconjugated monoclonal mouse anti-CD44 antibody (AbD Serotech) for 15 min at RT in the dark. After centrifugation, samples were incubated with secondary antibody goat anti-mouse Alexa 488 (Santa Cruz Biotechnology) for 15 min at RT. Data were analysed by a FACSCanto II flow cytometer (BD Biosciences) and expressed as the percentage (%) ± standard deviation (SD) of positive cells compared with the II antibody (Ab). Samples treated only with the secondary antibody were taken as controls. 

### 2.6. Western Blot

At 24 and 48 h after irradiation, nuclear and cytoplasmic proteins were obtained by means of the NER PER Nuclear and Cytoplasmic Extraction Reagents kit (Thermo Fisher Scientific), whereas at 4 h, histones were extracted with the Histone Extraction Kit—Rapid/Ultra-Pure (Abcam, Cambridge, UK), according to the manufacturer’s instructions. To detect phosphorylated histone H2AX, cultures treated with 0.1-μM 8-MOP photoactivated with 5-J/cm^2^ UVA light were used as the positive control [10]. Irradiation was carried out by using the Philips HPW 125 lamp (Philips Lighting Italy S.p.A., Milano, Italy), mainly emitted at 365 nm (radiant power about 8 mW/cm^2^). Protein quantification was carried out using the BCA Protein Assay Reagent Kit (Thermo Fisher Scientific) following the manufacturer’s protocols. Thirty micrograms of nuclear or cytoplasmic proteins were separated by 10% SDS/PAGE, whereas histones were separated with a 18% gel. Then, proteins were electrophoretically transferred onto polyvinylidene difluoride (PVDF) membranes (Immobilon PVDF, Millipore, Watford, UK). Blocking of nonspecific binding sites was achieved by incubation with 5% BSA in tris-buffered saline for 2 h at RT. The immunoblot was carried out by overnight incubation at 4 °C with primary polyclonal rabbit anti-human glyceraldehyde 3-phosphate dehydrogenase (GAPDH), histone H4 (Millipore, Burlington, MA, USA), phospho(Thr180/Tyr182)-p38, phospho(Thr202/Tyr204)-p44/42, phospho(Ser139)histone H2AX (Cell Signaling, Danvers, MA, USA), phospho(Tyr 216)GSK3β (Abcam, Cambridge, UK) (1:500, *v*/*v*), primary monoclonal mouse anti-human β-catenin, phospho(Ser 33/Ser 37)β-catenin, and laminB1 (Santa Cruz, Dallas, TX, USA) (1:500, *v*/*v*) antibodies. The detection of target proteins was performed using peroxidase-conjugated goat anti-rabbit (Bio-Rad Laboratories, Inc.) or anti-mouse secondary antibodies (Abcam). The development of immunoreactivity was enhanced by a chemiluminescence substrate (ECL; Bio-Rad) and then visualised by the VersaDoc Imaging System (Bio-Rad, Hercules, CA, USA). The protein expression level was normalised to housekeeping proteins GAPDH, laminB1, and H4 and quantified using the image processing software ImageJ.

### 2.7. Reverse Transcription Polymerase Chain Reaction (RT-PCR)

At 3 h from irradiation, total RNA was extracted using Trizol^®^ Reagent (Sigma-Aldrich) according to the manufacturer’s instructions and quantified using NANODROP 2000 (Thermo Fisher Scientific). Primers were obtained from Invitrogen, and their sequences are reported in Table 1. β-actin was chosen as the housekeeping gene. RT-PCR was carried out through the Qiagen^®^ One Step RT-PCR Kit (Qiagen, Crawley, UK) according to the manufacturer’s protocols and using total RNA at a concentration of 50 ng/reaction for each sample. The thermal cycling program consisted of 50 °C for 30 min (reverse transcription), 95 °C for 15 min (DNA-polymerase activation), 39 two-step cycles of 94 °C for 1 min (denaturation), 57 °C for 1 min (annealing), and 72 °C for 1 min (elongation). The procedure was carried out using the iCycler iQ™ (Bio-Rad). The PCR products were separated by 2% agarose gel electrophoresis and visualised by Gel Red Nucleic Acid staining at 1:10,000 (Biotium, Hayward, CA, USA). The images of the gel were captured with Gel Doc^TM^ Imager (Bio-Rad) and analysed with Image Lab software (Bio-Rad). To obtain a semiquantitative assessment of the gene expression, data were expressed as normalised ratios by comparing the integrated density values for target genes with those for β-actin.

### 2.8. Statistical Analysis

Data were expressed as mean ± the standard error of the mean. The difference between groups was evaluated using analysis of variance (ANOVA) and Student’s *t*-test.

## 3. Results

### 3.1. Antiproliferative Effects and Genotoxicity of TMA/BL

In all experiments, the photoactivation of TMA was achieved by using a BL light intensity of 2 J/cm^2^, which did not affect the cell growth; indeed, higher radiant exposures inhibited cell proliferation. Photoactivated TMA (TMA/BL) induced significant decreases in cell numbers compared to untreated cultures taken as the control, whereas the compound kept in the dark was ineffective (Figure 2a,b). Further experiments were carried out by treating cultures with 5-µM TMA/BL corresponding to about 50% cell growth inhibition. Figure 2a,b also reports the effects of 1-mM LiCl with and without 5-µM TMA/BL on the cell viability. LiCl, an inhibitor of GSK3β, was used to verify the involvement of this kinase in the effects of TMA/BL on Wnt signalling (see below). The antiproliferative activity of photoactivated TMA was due to a proapoptotic effect (Figure 2c,d). At both time points, the percentages of the apoptotic cells were significantly higher in TMA/BL-treated cultures in comparison with those detected in the control ones. At 24 h after treatment, significant increases in the amount of apoptotic annexin V-positive cells were also observed in cultures treated with only light or with the compound kept in the dark.

To evaluate the photogenotoxicity of TMA/BL, the phosphorylation status of histone H2AX was determined by Western blotting (WB). As shown in Figure 3, no variations were observed in cultures treated with TMA/BL, only BL or the compound kept in the dark. On the contrary, significant increases of the phosphorylated form of histone H2AX were detected in cultures treated with 8-MOP photoactivated by UVA light.

### 3.2. TMA/BL Affected the Canonical Wnt Pathway

At 24 and 48 h, significant decreases in the expression of nuclear β-catenin were detected in cultures treated with photoactivated TMA (Figure 4A,B). In the control cultures, the immunoreactivity towards β-catenin was well visible in both the nucleus and cytoplasm (Figure 4C). On the contrary, in TMA/BL-treated cells, the β-catenin-positive areas were reduced and completely absent in the nucleus. 

The modulation of nuclear β-catenin led to decreases in the expression levels of its target genes. At 3 h after the treatment with TMA/BL, cyclin D1, c-Myc, β-catenin, and CD44v6 mRNAs were significantly reduced compared to those detected in the control cultures (Figure 5). Notably, β-catenin, CD44v6, and c-Myc expressions were also inhibited by TMA kept in the dark.

Consistent with the results of the RT-PCR, TMA/BL significantly lowered the percentage of CD44-positive cells with respect to the untreated cultures (Figure 6a,b). At 24 and 48 h, 9% and 45% decreases were detected, respectively. The cytofluorimetric data were confirmed by immunofluorescence (Figure 6c). In untreated cultures, all cells showed a strong immunoreactivity towards CD44v6, mainly localised at the plasma membrane. On the contrary, only a few cells were slightly positive for CD44 in TMA/BL-treated cultures.

### 3.3. TMA/BL Affected the Canonical Wnt Pathway through the Modulation of GSK3β

The reduction of nuclear β-catenin was linked to an induction of its removal at the proteasomal level. At 24 and 48 h, significant increases in the phosphorylated forms of β-catenin and GSK3β kinases at tyrosine 216 were detected in TMA/BL-treated cultures (Figure 7). At 24 h, also, TMA kept in the dark raised the expression levels of these proteins.

To verify whether TMA/BL could affect the canonical Wnt pathway through the modulation of GSK3β, LiCl, an inhibitor of this kinase, was added to the TMA/BL-treated cultures. As reported in Figure 2, at 24 and 48 h, 1-mM LiCl alone increased the cell proliferation with respect to the controls and counteracted the decreases in cell viability induced by TMA/BL. The salt significantly lowered the expression of the phosphorylated forms of both β-catenin (24 and 48 h) and GSK3β (24 h) with respect to those determined in the TMA/BL-treated cultures without restoring the expression values to those of the untreated cultures (Figure 7). Furthermore, at 24 and 48 h, it enhanced the nuclear β-catenin and CD44 expressions, respectively (Figure 4).

### 3.4. TMA/BL Affected some Mitogen-Activated Protein Kinases (MAPKs)

Finally, the activation statuses of some signalling pathways involved in cell proliferation and apoptosis and overexpressed in cancer cells were verified by Western blotting (Figure 8). At 24 and 48 h after the treatment with TMA/BL, the expression levels of phosphorylated extracellular signal-regulated kinase (ERK) ½ decreased, whereas phosphorylated p38 was enhanced at 48 h. Notably, also, the cultures treated with the compound kept in the dark presented a lower expression of p-ERK ½ than the control ones. No significant variations of the phosphorylated form of Akt were detected at either time point (data not shown).

## 4. Discussion

In the attempt to improve the photocytotoxic activity of psoralens and avoid severe side effects, which are mainly related to the formation of XLs between the DNA strands under UVA light [11], TMA can offer the advantage to be cytotoxic under both UVA and BL, behaving as a monofunctional compound under BL.

The reliable photoactivity of TMA arises from its extended aromatic structure, typical of furocoumarins, that enhances its ability to absorb UVA radiation. Indeed, the presence of three methyl groups gives to TMA the suitable lipophilicity for a strong intercalation inside the DNA, and, at the same time, their position enhances the photoreactivity of the two double bonds towards the pyrimidine bases. Moreover, a tail of absorption in the near-visible region enables TMA to photoreact under blue wavelengths, as for the linear analogue 8-MOP [9,12].

Our data showed that TMA activated by BL behaved as a strong antiproliferative compound on T24 human bladder cancer cells through the induction of apoptosis.

In these cells, TMA enhanced the phosphorylation status of p38, whose signalling pathway modulates cell cycle checkpoints, cell apoptosis, and autophagy upon the effects of different extracellular stimuli, i.e., oxidative stress and light [13]. On the opposite end, the photoactivated TMA decreased the phosphorylation of the p44/42 MAP kinases (ERK ½) that are usually activated by growth factors and mitogens, thus leading to antiapoptotic effects by the downregulation of proapoptotic factors and/or upregulation of antiapoptotic ones [14,15].

TMA/BL was also able to decrease the expression of c-Myc that promotes cell proliferation but also sensitises cells to various extracellular apoptotic stimuli [16,17,18].

The same effect was obtained on the expression of cyclin D1, whose high levels are related to a poor prognosis and tumour recurrence in many cancers, including bladder cancer [19].

Furthermore, the photoactivation of TMA by BL affected the canonical Wnt signalling pathway in T24 cells through modifications of the β-catenin and GSK3β expression levels, whose dysregulations are associated with several types of cancers [20]. Indeed, the Wnt/β-cat signalling pathway plays varied roles in intercellular adhesion, cellular differentiation and proliferation, and stem cell maintenance, as well as in disease states such as cancer. The nuclear expression of β-catenin in T24 cells was strongly reduced by the TMA/BL treatment. As for other tumours, the reduction and loss of β-catenin might disrupt the stability and integrity of the E-cadherin–catenin complex and impair the cellular adhesive junction, inhibiting cell proliferation, invasion, and metastasis [21].

Additionally, glycogen synthase kinase 3β (GSK3β), whose role in glucose homeostasis has been extensively studied [22], participates in many cellular processes [23], and its dysregulation has been implicated in a wide range of diseases, such as obesity, type 2 diabetes, Alzheimer disease, and cancer [24,25]. The inactivation of GSK3β by phosphorylation at specific residues is a primary mechanism by which this constitutively active kinase is controlled, and the effect induced by photoactivated TMA on the modulation of GSK3β of T24 cells may predict the anticancer activity induction by the proposed TMA/BL treatment. The introduction of LiCl in our experiments and its role as a classical GSK3β inhibitor confirmed the involvement of this target in the TMA/BL effect on T24 cells [26].

Remarkably, the decrease in nuclear β-catenin was associated with a lower percentage of cells presenting the CD44 cell surface receptor, a transmembrane glycoprotein that plays an important role in cell adhesion, migration, and proliferation [27]. As a matter of fact, CD44 has been shown to be overexpressed in various types of cancers, and a subfamily of CD44 splice variants encompassing the variant domain 6 (CD44v6) has been implicated in the metastatic potential of tumours [28,29].

The effect of lowering the CD44v6 expression in T24 cells suggests the potential suppression of the proliferation and invasiveness of bladder carcinoma by photoactivated TMA treatment. It must be noted that CD44 protects CSCs from the cytotoxic effects of ROS [30] by promoting the synthesis of glutathione and activating the glycolytic pathway. Thus, the targeting of CD44 can also increase the sensitivity of CSCs to ROS, improving their eradication. Interestingly, decreases in the percentage of CD44-positive cells were also detected in cultures treated with the compound kept in the dark, suggesting that TMA could also interact with molecular targets without photoactivation.

Interestingly, TMA did not exert photogenotoxicity on T24 cells in comparison with 8-methoxypsoralen (8-MOP), as already reported in previous studies on methyl angelicins as potential photochemotherapeutic agents [31]. The lower degree of H2AX histone phosphorylation induced by TMA compared to 8-MOP gives additional proof that the DNA damage induced by TMA under BL is different from that of 8-MOP [32].

## 5. Conclusions

In conclusion, TMA photoactivated by BL may represent an interesting option for the photochemotherapy of noninvasive bladder carcinomas, because this treatment is able to inhibit the key pathways for tumour growth and progression in the absence of genotoxic effects. As BL is slightly more penetrating than UVA, the application of TMA in (superficial) an internal tumour, such as bladder cancer, can be envisaged. This could be done with the help of an optical fibre emitting blue light, able to reach the diseased area and to activate TMA, previously administered through, for example, a catheter balloon.

## Figures and Tables

**Figure 1 biomolecules-11-00158-f001:**
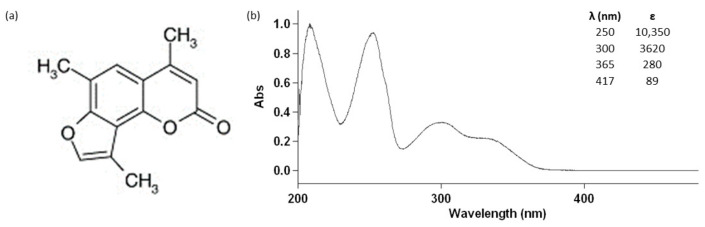
Chemical structure (**a**) and absorption spectrum (**b**) of 4,6,4′-trimethylangelicin (TMA) and its molar extinction coefficients at 250 nm, 300 nm, 365 nm, and 417 nm.

**Figure 2 biomolecules-11-00158-f002:**
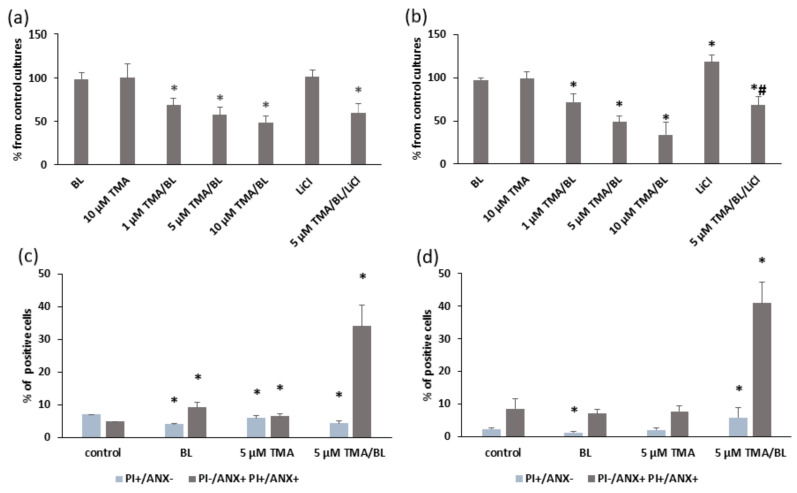
Cell viability (**a**,**b**) and apoptosis (**c**,**d**) determined by trypan blue staining and a FACS analysis with annexin V (ANX) and propidium iodide (PI), respectively, at 24 (**a**,**c**) and 48 (**b**,**d**) h after the treatment with TMA, blue light (BL) (2 J/cm^2^), and 1-mM LiCl. Data were obtained from 3 experiments carried out in triplicate and reported as mean ± SD. * = *p* < 0.05 vs. untreated cultures taken as the control and # = *p* < 0.05 vs. 5-µM TMA/BL-treated cells, Student’s *t*-test.

**Figure 3 biomolecules-11-00158-f003:**
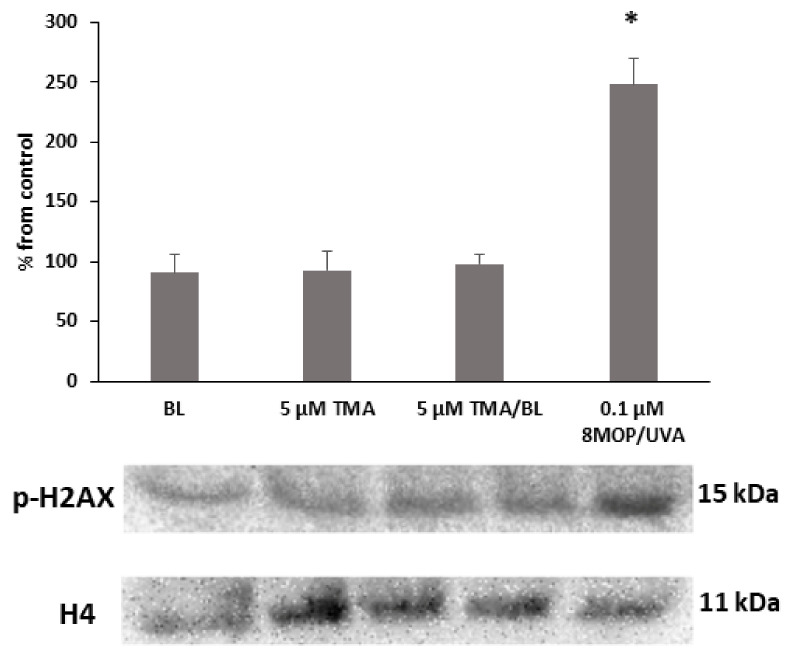
Western blot analysis of the phosphorylated form of the histone H2AX. The cultures were treated with 5-µM TMA/BL (2 J/cm^2^). Quantification of the expression levels was performed by a densitometric analysis using ImageJ software. Data, normalised to the H4 histone, are reported as percentages from the control cultures. * *p* < 0.05 vs. control. 8-MOP: 8-methoxypsoralen.

**Figure 4 biomolecules-11-00158-f004:**
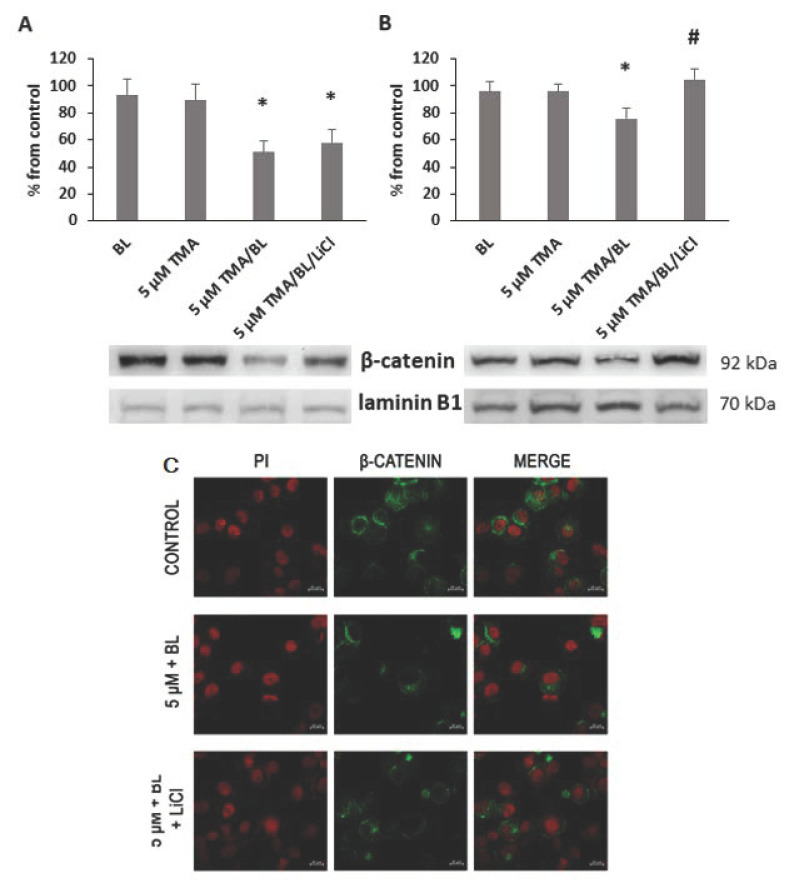
Nuclear expression of β-catenin. At 24 (**A**) and 48 (**B**,**C**) h after the treatment with TMA, BL (2 J/cm^2^), and 1-mM LiCl, the β-catenin expression was verified by Western blotting (WB) (**A**,**B**) and immunofluorescence (**C**). Quantification of the expression levels was carried out by a densitometric analysis using ImageJ software (**A**,**B**). Data were reported as percentages from the control cultures. * = *p* < 0.05 vs. control cultures and # = *p* < 0.05 vs. cultures treated with TMA/BL, Student’s *t*-test. (**c**) Nuclei and β-catenin-positive areas are in red and green fluorescence, respectively (bar: 20 μm).

**Figure 5 biomolecules-11-00158-f005:**
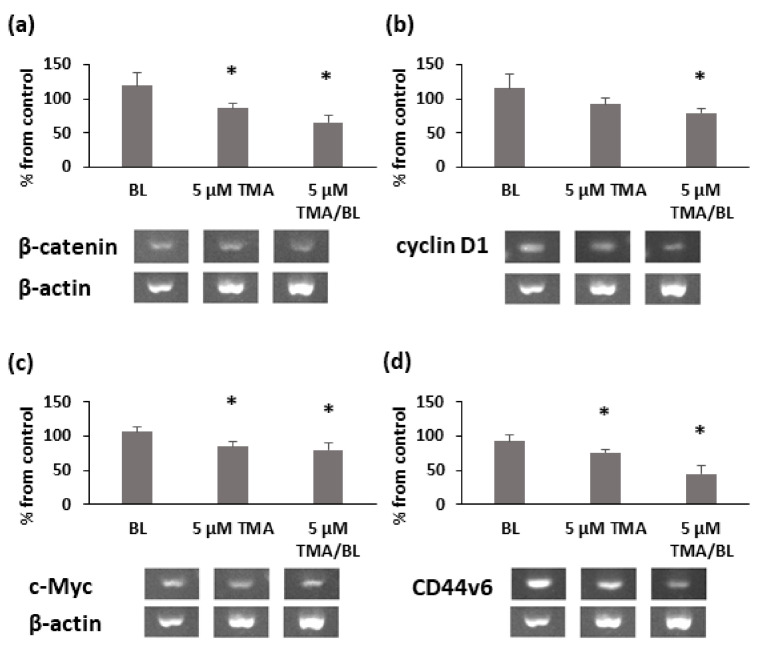
RT-PCR analysis of β-catenin (**a**), cyclin D1 (**b**), c-Myc (**c**), and CD44v6 (**d**) at 3h after the treatment with TMA/BL (2 J/cm^2^). Quantification of the expression levels was carried out by a densitometric analysis using ImageLab software. Data were reported as percentages from the control cultures. * = *p* < 0.05 vs. control cultures, Student’s *t*-test.

**Figure 6 biomolecules-11-00158-f006:**
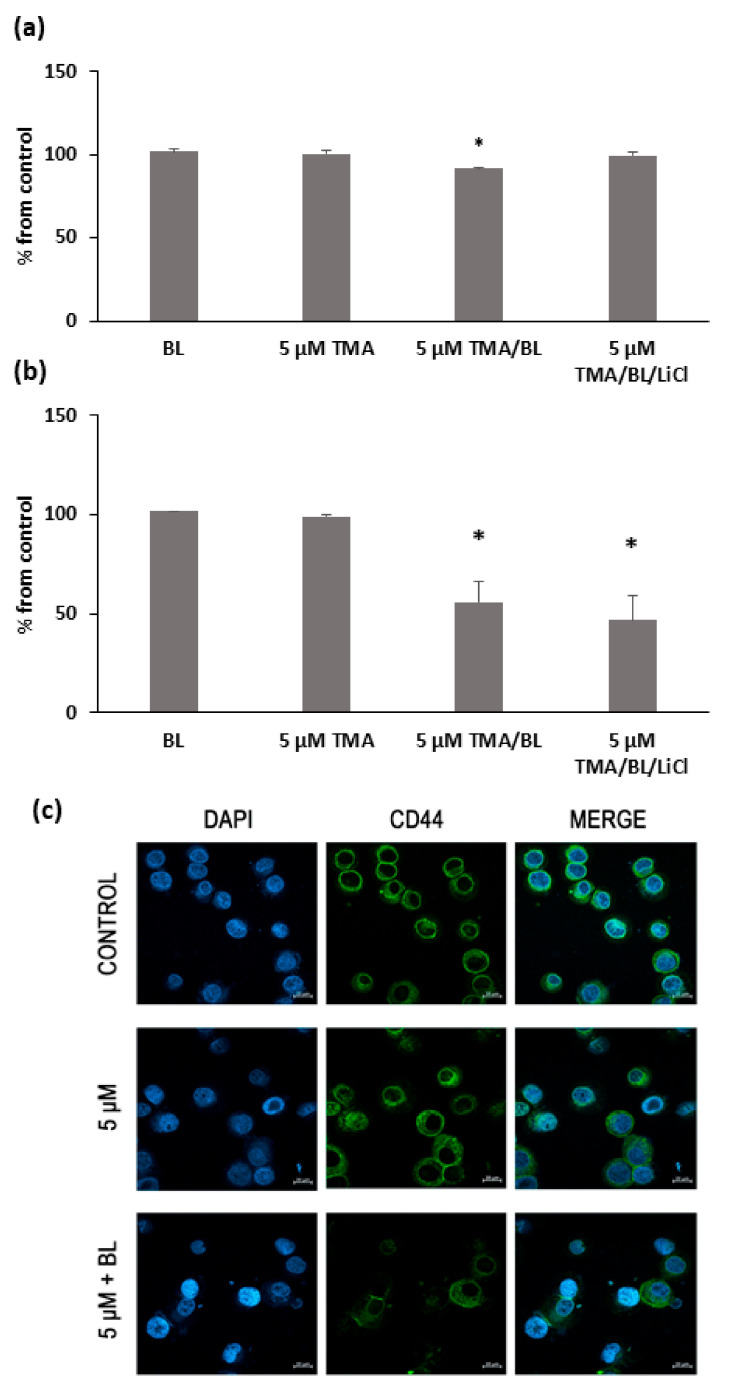
FACS analysis of the CD44 expression at 24 (**a**) and 48 h (**b**) after the treatment with TMA, BL (2 J/cm^2^), and 1-mM LiCl. Data are reported as % of positive cells compared to the control cultures. The results are expressed as percentages from the control cultures. * = *p* < 0.05 vs. the control cultures. (**c**) Immunofluorescence with anti-CD44 antibody at 24 h after the treatment with TMA/BL (2 J/cm^2^). Nuclei and CD44-positive areas are in blue and green fluorescence, respectively (bar: 20 μm).

**Figure 7 biomolecules-11-00158-f007:**
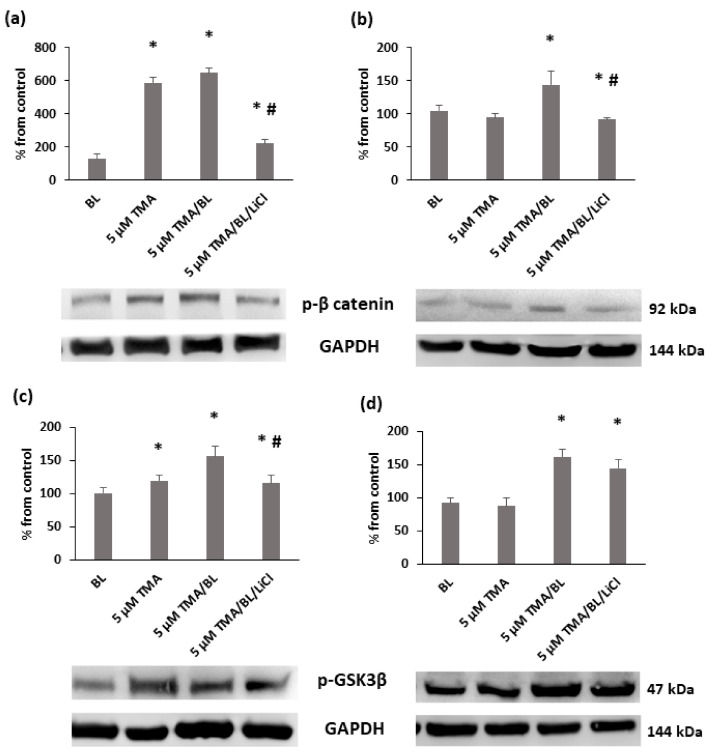
Western blot analysis of the phosphorylated forms of β-catenin (**a**,**b**) e GSK3β (**c**,**d**) at 24 (**a**,**c**) and 48 h (**b**,**d**) after the treatment with TMA, BL (2 J/cm^2^), and 1-mM LiCl. Quantification of the expression levels was carried out by a densitometric analysis using ImageJ software. Data are reported as percentages from the control cultures. * = *p* < 0.05 vs. the control cultures, # = *p* < 0.05 vs. the cultures treated with TMA/BL, Student’s *t*-test.

**Figure 8 biomolecules-11-00158-f008:**
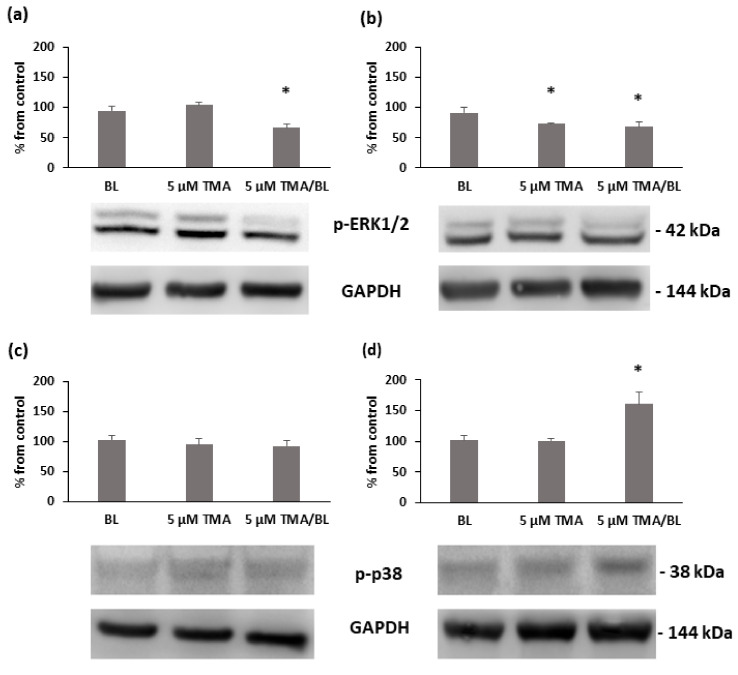
Expression of the phosphorylated forms of ERK 1/2 (**a**,**b**) and p38 (**c**,**d**) at 24 (**a**,**c**) and 48 (**b**,**d**) h after the treatment with TMA/BL (2 J/cm^2^). Quantification of the protein expression levels was performed by a densitometric analysis using ImageJ software, and the results are expressed as percentages of the control cultures * = *p* < 0.05 vs. the control cultures, Student’s *t*-test.

**Table 1 biomolecules-11-00158-t001:** Primers for PCR amplification.

Gene	Primer Sequence	Accession
β-Actin	F-TGACGTGGACATCCGCAAAG R-TGGAAGGTGGACAGCGAGG	NM_001101
β-Catenin	F-CTTCACCTGACAGATCCAAGTC R-CCTTCCATCCCTTCCTGTTTAG	NM_001904
cyclin D1	F-GCGGAGGAGAACAAACAGAT R-GAGGGCGGATTGGAAATGA	NM_053056.2
c-myc	F-CTCCACACATCAGCACAACTA R-TGTCCAACTTGACCCTCTTG	NM_002467
CD44v6	F-TCCAGGCAACTCCTA R-CAGCTGTCCCTGTTG	NM_001202555.1

## Data Availability

The authors confirm that the data supporting the findings of this study are available within the article.

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
