# Peer review of "4,6,4′-Trimethylangelicin Photoactivated by Blue Light Might Represent an Interesting Option for Photochemotherapy of Non-Invasive Bladder Carcinoma: An In Vitro Study on T24 Cells"

_biomolecules, 2021, doi:10.3390/biom11020158_

Round 1

Reviewer 1 Report

The authors evaluated the photodynamic effect of TMA on human bladder cancer cells and explored the underlying mechanisms in its cytotoxic action. Photoactivated TMA induced antiproliferative activity via induction of apoptosis without genotoxicity. Nevertheless, there are several major issues that must be addressed in order to the manuscript be suitable for publication:

  1. In the abstract, I suggest to include what are H2A, GSK3β and CD44v6.
  2. Regarding the methods section, it is not clear how did the authors perform the incubation with the TMA. The incubation of the PS was in PBS or medium? The excess of TMA (that did not enter the cells) was removed before irradiation?
  3. I suggest that the authors expose earlier the rational for the use of LiCl.
  4. In page 4, the authors refer “whereas the compound kept in dark was ineffective (fig. 2 a and b), I suppose that the authors meant fig.1 a and b.
  5. In the graphs of all figures, good labels are lacking. I strongly suggest that the authors include Y axis labels, with identification which variable are measuring.
  6. In Western blots, it would be important show the molecular weights protein bands.
  7. In page 6, the authors refer “At 24 and 48 h, significant decreases in the expression of nuclear β-catenin” but they performed an analysis of the protein levels, instead of mRNA expression levels. In addition, it seems that the decrease in b-catenin protein levels after 48h is not so evident as for 24h (in TMS/BL). What could be the reason for that?
  8. Concerning the RT-PCR data (fig. 4), I suggest to change the order of the images b-actin and each of the other genes, always keeping b -actin below the other genes.
  9. The authors should show the immunoreactivity of CD44 when cells were incubated with TMA (in the absence of blue light).
  10. In fig. 6 c, the western blot of p-GSK3b should be replaced by a more representative picture of the quantitative data.

Author Response

  1. In the abstract, I suggest to include what are H2A, GSK3β and CD44v6.

All the required information have been included in the Abstract: “……phospho-H2AX, an indicator of DNA double strand breaks…”, “ …. GSK3β phosphorylation state (Tyr 216), that induces proteasomal degradation of β-catenin…”, and “…CD44v6, a marker of cancer stem cells...”

  1. Regarding the methods section, it is not clear how did the authors perform the incubation with the TMA. The incubation of the PS was in PBS or medium? The excess of TMA (that did not enter the cells) was removed before irradiation?

The incubation with TMA was carried out in MEM medium w/o phenol red, as already reported in the text: line “… and replaced with MEM w/o phenol red (Sigma-Aldrich) containing various concentration of TMA…” (line 83-84).

The excess of TMA (that did not enter the cells) was not removed before irradiation, but after irradiation, as indicated in the manuscript: “The samples were maintained at room temperature (RT) during irradiation. Then, media were removed and replaced with the proliferative one.” (line 91-92)

  1. I suggest that the authors expose earlier the rational for the use of LiCl.

Accordingly, in point 3.1 the following sentences have been added: Figures 2a and 2b also report the effects of 1 mM LiCl with and without 5 µM TMA/BL on cell viabililty. LiCl, an inhibitor of GSK3β, was used to verify the involvement of this kinase in the effects of TMA/BL on Wnt signalling (see below).” Furthermore, the legends of figures 2, 3, 6, and 7 (ex 1, 2, 5, and 6) have been modified by adding LiCl concentration.

  1. In page 4, the authors refer “whereas the compound kept in dark was ineffective (fig. 2 a and b), I suppose that the authors meant fig.1 a and b.

We are so sorry for the mistake. Since we have added one figure (Figure 1) in the Introduction section showing the chemical structure of TMA (as requested from Reviewer 2), the number of all figures has been modified and figure 2 remains figure 2.

  1. In the graphs of all figures, good labels are lacking. I strongly suggest that the authors include Y axis labels, with identification which variable are measuring.

We have not understood this suggestion. In all graphs, Y axis reports the numerical scale (from 0 to 50/120/150/200) and the indication of the measured variable (% of positive cells or % from control where controls are non-treated cultures, as reported in the corresponding legends).

  1. In Western blots, it would be important show the molecular weights protein bands.

As suggested, figures have been modified by adding the molecular weights of protein bands.

  1. In page 6, the authors refer “At 24 and 48 h, significant decreases in the expression of nuclear β-catenin” but they performed an analysis of the protein levels, instead of mRNA expression levels. In addition, it seems that the decrease in b-catenin protein levels after 48h is not so evident as for 24h (in TMS/BL). What could be the reason for that?

The transcript levels of β-catenin have been also measured, as already reported in figure 5 (previously figure 4).  It is true that the decreases in b-catenin protein levels after 48h were not so evident as for 24 h. We can suppose that cells can recover Wnt signalling over the time.

  1. Concerning the RT-PCR data (fig. 4), I suggest to change the order of the images b-actin and each of the other genes, always keeping b -actin below the other genes.

We thank the reviewer for the useful suggestion. Accordingly, figure 5 (previously figure 4) has been modified.

  1. The authors should show the immunoreactivity of CD44 when cells were incubated with TMA (in the absence of blue light).

We thank the reviewer for the useful suggestion. Accordingly, figure 6 (previously figure 5) has been modified.

  1. In fig. 6 c, the western blot of p-GSK3b should be replaced by a more representative picture of the quantitative data.

We are so sorry because we have not understood this suggestion. Figure 7c (previously figure 6c) reports the quantification of expression levels by densitometric analysis in the same manner of figures 7a, 7b, and 7d (previously figure 6a, 6b, and 6d).

Reviewer 2 Report

The manuscript by Miolo et al. describes a rather careful piece of work that elegantly demonstrates that blue-light activated angelicin inactivate T24 bladder cancer cells without photogenotoxic effects. The work reinforces the authors’ previous findings on DU145 prostate cancer cells and helps building a strong case for angelicin/blue light as a novel weapon against cancer. This line of research is worth pursuing and assessed in in vivo models. The paper is well written, the conclusions are very well supported by the comprehensive results presented and, overall, the manuscript is an example of good science. I invite the authors to consider a few suggestions to improve the readability of their already excellent manuscript.

1) For the benefit of the readers outside of the psoralens community, consider adding a figure with the structure and absorption spectrum of angelicin to better appreciate the differences between UVA- and blue-light irradiation.

2) Line 193. This paragraph is a bit confusing. Is there any difference between photoactivated TMA and TMA/BL? If not, please unify the nomenclature throughout the manuscript.

3) Some of the results shown in figures 1, 3, 5 and 6 were obtained in the presence of LiCl but neither the caption nor the nearby text provides any explanation for this. The reason is not given until line 236, where we learn that LiCl is a GSK3β inhibitor. It would facilitate the reading of the text if that information would be given the first time that LiCl results are shown.

Author Response

  1. For the benefit of the readers outside of the psoralens community, consider adding a figure with the structure and absorption spectrum of angelicin to better appreciate the differences between UVA- and blue-light irradiation.

The requested figure has been added at the end of the Introduction section (figure 1). Consequently, the number of the other figures has been modified.

  1. Line 193. This paragraph is a bit confusing. Is there any difference between photoactivated TMA and TMA/BL? If not, please unify the nomenclature throughout the manuscript.

There is no difference between photoactivated TMA and TMA/BL. As suggested, we have specified this in the Introduction (line 75) and Results (line 172) sections.

  1. Some of the results shown in figures 1, 3, 5 and 6 were obtained in the presence of LiCl but neither the caption nor the nearby text provides any explanation for this. The reason is not given until line 236, where we learn that LiCl is a GSK3β It would facilitate the reading of the text if that information would be given the first time that LiCl results are shown.

We thank the reviewer for the useful suggestion. Accordingly, in point 3.1 the following sentences have been added: Figures 2a and 2b also report the effects of 1 mM LiCl with and without 5 µM TMA/BL on cell viabililty. LiCl, an inhibitor of GSK3β, was used to verify the involvement of this kinase in the effects of TMA/BL on Wnt signalling (see below).” Furthermore, the legends of figures 2, 3, 6, and 7 (ex 1, 2, 5, and 6) have been modified by adding LiCl concentration.

Reviewer 3 Report

I find it a very interesting article, as it opens up another path of treatment for bladder cancer. I have a doubt, only one irradiation was done?what would happen if more were done?
In clinical practice, they would probably require more than one photodynamic therapy session. Since we know that for non-melanoma skin cancer, multiple sessions are required to obtain a satisfactory result. Perhaps I miss this fact.

How many human bladder cancer were cultured?an controls?

Author Response

  1. I have a doubt, only one irradiation was done? what would happen if more were done?

We have chosen 2J/cm2 as light dose because it did not exert any toxic effects on T24 cells. Higher doses significantly impaired cell viability and, consequently, they were not taken in account to avoid bias in the evaluation of antiproliferative activity of photoactivated TMA (TMA/BL).

  1. In clinical practice, they would probably require more than one photodynamic therapy session. Since we know that for non-melanoma skin cancer, multiple sessions are required to obtain a satisfactory result. Perhaps I miss this fact.

We agree to the reviewer. Indeed, multi-irradiation modality is used in photodynamic therapy and adapted because of the in vivo clinical response. However, in our work in vitro experiments did not require multiple light doses as a unique irradiation step provided good antiproliferative responses.

  1. How many human bladder cancer were cultured? an controls?

Only T24 cells were used; seeding density (5x104/35mm Petri dish) has been reported in materials and Methods section (line 85). Sorry, I don’t understand what do you mean for “an controls?”.